# Inflammatory Cytokines, Redox Status, and Cardiovascular Diseases Risk after Weight Loss via Bariatric Surgery and Lifestyle Intervention

**DOI:** 10.3390/medicina59040751

**Published:** 2023-04-12

**Authors:** Mahmoud M. A. Abulmeaty, Hazem K. Ghneim, Abdulaziz Alkhathaami, Khalid Alnumair, Mohamed Al Zaben, Suhail Razak, Yazeed A. Al-Sheikh

**Affiliations:** 1Community Health Sciences Department, College of Applied Medical Sciences, King Saud University, Riyadh 11433, Saudi Arabia; aalkhathaami@ksu.edu.sa (A.A.); alnumair@ksu.edu.sa (K.A.); smarazi@ksu.edu.sa (S.R.); 2Department of Clinical Laboratory Sciences, College of Applied Medical Sciences, King Saud University, Riyadh 11362, Saudi Arabia; hghneim@ksu.edu.sa (H.K.G.);; 3Surgery Department, Sultan Bin Abdulaziz Humanitarian City, Riyadh 13571, Saudi Arabia; malzaben@sbahc.org.sa

**Keywords:** inflammatory cytokines, cardiovascular diseases risk, bariatric surgery, lifestyle intervention, weight loss

## Abstract

*Background and Objectives*: Obesity is a chronic inflammatory condition and is considered a major risk factor for cardiovascular disease (CVD). The effects of obesity management via sleeve gastrectomy (SG) and lifestyle intervention (LS) on inflammatory cytokines, redox status, and CVD risk were studied in this work. *Materials and Methods*: A total of 92 participants (18 to 60 years old) with obesity (BMI ≥ 35 kg/m^2^ were divided into two groups: the bariatric surgery (BS) group (*n* = 30), and the LS group (*n* = 62). According to the achievement of 7% weight loss after 6 months, the participants were allocated to either the BS group, the weight loss (WL) group, or the weight resistance (WR) group. Assessments were performed for body composition (by bioelectric impedance), inflammatory markers (by ELISA kits), oxidative stress (OS), antioxidants (by spectrophotometry), and CVD risk (by the Framingham risk score (FRS) and lifetime atherosclerotic cardiovascular disease risk (ASCVD)). Measurements were taken before and after six months of either SG or LS (500 kcal deficit balanced diet, physical activity, and behavioral modification). *Results*: At the final assessment, only 18 participants in the BS group, 14 participants in the WL group, and 24 participants in the WR group remained. The loss in fat mass (FM) and weight loss were greatest in the BS group (*p* < 0.0001). Levels of IL-6, TNF-a, MCP-1, CRP, and OS indicators were significantly reduced in the BS and WL groups. The WR group had significant change only in MCP-1 and CRP. Significant reductions in the CVD risk in the WL and BS groups were detected only when using FRS rather than ASCVD. The FM loss correlated inversely with FRS-BMI and ASCVD in the BS group, whereas in the WL group, FM loss correlated only with ASCVD. *Conclusions*: BS produced superior weight and fat mass loss. However, both BS and LS produced a similar reduction in the inflammatory cytokines, relief of OS indicators, and enhancement of antioxidant capacity, and consequently reduced the CVD risk.

## 1. Introduction

Obesity is an excessive accumulation of body fats due to a state of positive energy balance. The adipose tissues (AT) release inflammatory mediators such as tumor necrosis factor and several interleukins. These cytokines are overexpressed in obesity [1]. Obesity is a major risk factor for cardiovascular disease (CVD). The mechanisms linking obesity and CVD are based on the expansion of the AT, which increases the occurrence of the dysregulated release of adipocytokines, hypoxia, AT inflammation, and impaired mitochondrial function. In addition, obesity predisposes to insulin resistance, abnormal lipid and glucose metabolism, hypertension, a pro-inflammatory state, and endothelial dysfunction [2]. The Framingham Risk Score (FRS) is a gender-specific algorithm that is used to estimate the CVD risk of an individual over the next ten years by using independent variables such as total cholesterol, systolic blood pressure, age, gender, high-density lipoprotein (HDL), and diabetes history [3]. FRS is the most commonly used method for predicting the risk of developing CVD on a long-term basis in patients with or without obesity [4].

The most effective method to reduce weight is bariatric surgery (BS). However, lifestyle intervention is the fundamental step and produces some significant results [5]. Sleeve gastrectomy is the most commonly used bariatric surgery in Saudi Arabia and worldwide. In sleeve gastrectomy, approximately 80% of the stomach is reduced and converted into a tubular-shaped stomach, restricting food intake and thereby leading to weight loss [6]. Lifestyle intervention procedures are not invasive and do not involve gastrointestinal anatomical changes. Although it results in significantly less weight loss, it does not entail long-term risks [5]. The eventual advantage of weight loss is that it is related to a decrease in obesity co-morbidities, an increase in the quality of life, and a decrease in CVD risk [7,8]. There is considerable evidence that laparoscopic sleeve gastrectomy could be used as a treatment for the reduction of CVD risks. Studies have shown that BS is an effective treatment for rapid weight loss and has encouraging effects on the decrease in ten-year CVD risk [9]. Amelioration of inflammation and oxidative stress are suggested mechanisms. Studies on the levels of IL-6, TNF-α, MCP-1, and other inflammatory cytokines after BS have been inconsistent. Some reports have stated reductions in the levels of these inflammatory cytokines, while others have reported no such reduction [10]. After BS, IL-6, CRP, and calprotectin levels were reduced together with a reduction in BMI (body mass index) [11]. Levels of oxidative stress markers were also reduced after gastric bypass surgery (RYGB) [12].

Non-surgical approaches to weight loss have been reported to produce successful improvement in the inflammatory, redox, and metabolic profiles in patients with obesity and metabolic syndrome [13]. Recently some evidence was collected about the value of lifestyle modifications (including physical activity and diet) in the improvement of abdominal obesity and some inflammation markers among overweight/obese patients with metabolic syndrome [14]. Amelioration of oxidative stress in form of a reduction in plasma levels of monocyte chemoattractant protein-1 (MCP1), xanthine oxidase, malondialdehyde (MDA), and myeloperoxidase is an additional benefit of long-term lifestyle intervention [15]. This indicates that oxidative stress has strong associations with biomarkers of cardiometabolic health [16]. The literature has few studies prospectively comparing BS versus lifestyle intervention regarding effects on oxidative, inflammatory, and cardiovascular disease risk in patients with obesity, especially in the Arabic region. Accordingly, this study examined, in a prospective manner, the parameters of oxidative stress, antioxidants, inflammation, and risks of CVD in patients undergoing weight loss by sleeve gastrectomy (SG) and lifestyle intervention (LS). 

## 2. Materials and Methods

### 2.1. Study Subjects

A total of 92 participants were non-randomly assigned into two cohorts: bariatric surgery (BS group n = 30) where participants underwent bariatric surgery (sleeve gastrectomy), and lifestyle intervention (LS group, n = 62) where participants followed multimodal lifestyle intervention (500 kcal deficit balanced diets, gradual physical activity, and behavioral modification). According to the participants’ weight loss achievement in the LS group, they were further divided into: (1) the weight loss (WL) group, in which participants succeeded in losing ≥7% of their initial weight; and (2) the weight resistant (WR) group, in which participants followed lifestyle modification, but no or mild weight changes occurred (<7% of their initial weight) (Figure 1). The inclusion criteria were that the participants were from 18 to 60 years of age and had BMI ≥ 40 kg/m^2^, or BMI ≥ 35 kg/m^2^ with co-morbidities [17]. Pregnancy, previous surgeries, physical impairment, mental disorders, taking drugs (including weight loss medications), and alcohol use were exclusions. The study protocol was approved by the IRB committee of the research center at Sultan Bin Abdulaziz Humanitarian City (SBAHC) under reference number 49-2021-IRB dated 03/06/2021.

### 2.2. Interventions 

In the BS group, laparoscopic sleeve gastrectomy was performed under general anesthesia by the same surgeon in the SBAHC. Routine preoperative, and postoperative care (including early post-operative dietary management) were performed. Baseline assessment was taken within the last week before the surgery during the preoperative preparation and the final assessment was recorded about 6 months after the surgery date. 

For participants in the lifestyle group, a baseline assessment was performed at the first visit, then every patient in the LS group was given a balanced 500–1000-calorie-deficit diet, a personalized physical activity plan, and customized behavioral modification according to our previous publication [18]. This Three-pyramid program for lifestyle intervention includes three domains of intervention. The first was a 500–1000-calorie-deficit balanced diet according to the USDA food guide pyramid. This mandates the calculation of an individual’s caloric intake and then the prescription of a diet containing 500 to 1000 fewer calories that follows the principles of the USDA food guide pyramid. The second was a gradual pyramid-like physical activity prescription. This mandates assessment of the individual’s physical activity status then adding sessions of moderate intensity physical activity (such as swimming, walking, etc.) for 60 min, 5 days/week in a gradual manner. The third domain was a pyramid-like model for behavioral modification to reach a healthier lifestyle. When interviewing of the participant, selected bad behaviors were scheduled for gradual change. The potential and barriers to change were discussed with the participant to find possible solutions. Subjects were asked to do self-monitoring even for smaller success. Discussion of relapse causes (if present) and continuous encouragement were essential elements. The final assessment was scheduled after 6 months. The participants who succeeded in losing >7% of the baseline weight were assigned to the weight loss (WL) group, while those who failed to reach this target were assigned to the weight resistant (WR) group. 

### 2.3. Anthropometric, Body Composition Changes

Weight, height, and body mass index were recorded. Body composition was assessed by a bioelectric impedance analyzer (Tanita BC-418, Tanita Corp., Tokyo, Japan). The parameters used for analysis included percent body fat (PBF), fat mass (FM), fat-free mass (FFM), fat mass index (FMI = FM/height^2^), fat-free mass index (FFMI = FFM/height^2^) [19], visceral fat (VF) rating, and total body water (TBW). The amount of weight loss was calculated as follows: weight loss = baseline weight–final weight. Additionally, fat mass loss was calculated by subtracting the final FM from the baseline FM.

### 2.4. Inflammatory Cytokines Assessment

Fasting blood samples were taken and serum was prepared and stored in a −80 freezer until the time of analyses. ELISA (Enzyme-linked Immunosorbent Assay) tests were performed for each of the inflammatory markers IL-6, TNF-α, and MCP-1 (R&D Systems, Inc., Minneapolis, MN, USA: catalog numbers; D6050, DTA00D, and DCP00, respectively), in addition to the C-reactive protein (MyBiosource, San Diego, CA, USA; catalog no: MBS2021863).

### 2.5. Oxidative Stress Markers and Antioxidants Assessment

All key antioxidant enzyme activities including glutathione S-transferase (GST), glutamate cysteine ligase (GCL), glutathione synthetase (GS), Glutathione peroxidase 1 (GPx), glutathione reductase (GR), superoxide dismutase (SOD), and catalase (CAT), were measured in serum samples using spectrophotometric assays. In addition, OS indicators such as hydrogen peroxide (H_2_O_2_), superoxide anions (SA), lipid peroxides (LPO), t-protein carbonyl content (PCC), peroxynitrite radical (PON), and nitric oxide (NO) as well as reduced and oxidized glutathione (GSH and GSSG), and the GSH/GSSG ratio were also assayed using spectrophotometric methodologies. Extensive details of all procedures were previously published by our group [20,21].

### 2.6. CVD Risk Assessment 

Age, gender, systolic blood pressure, BMI, lipid panel, smoking, presence of diabetes, and presence of hypertension were used for estimating the CVD risk according to four FRS algorithms: (a) FRS based on BMI for hard cardiovascular disease which includes acute myocardial infarction, death due to coronary cause, and stroke (FRS-BMI-Hard CVD); (b) FRS based on the lipid panel for hard cardiovascular disease (FRS-Lipid-Hard CVD); (c) FRS based on BMI for full CVD, including hard CVD or other events such as coronary insufficiency, angina pectoris, and transient ischemic attack (FRS-BMI-Full CVD); and (d) FRS based on the lipid panel for full cardiovascular disease (FRS-Lipid-Full CVD). Furthermore, the lifetime atherosclerotic cardiovascular disease risk (ASCVD) was calculated according to the American Heart Association [19,22]. 

### 2.7. Statistical Analysis

The Shapiro–Wilk test was used to test if the study variables follow a normal distribution or not. The study parameters were compared among study groups by one-way ANOVA and Tukey’s HSD test for post hoc analysis or the Kruskal–Wallis H test with the intergroup comparisons with the Mann–Whitney U test. Repeated measurements were assessed by paired *t*-test or the Wilcoxon test. *p*-values < 0.05 were considered statistically significant. Pearson or Spearman correlations were used to test the association between weight loss and changes in inflammatory parameters and CVD risk. The SPSS software version 25 (SPSS Inc., Chicago, IL, USA) was used for all analyses.

## 3. Results

### 3.1. Baseline Characteristic Data among Study Groups

At the baseline assessment, the main characteristics were similar for the BS and LS cohorts, as shown in Table 1. Females represented 44% of the BS group and 46% of the LS group. 

### 3.2. Final Characteristic Data among Study Groups

At the final assessment, committed participants in the LS group were divided into WL (n = 14, 42.8% women) and WR (n = 24, 41.7% women). The remaining participants had dropped out (n = 24). The BS group also had a no-show for 12 participants. All attending participants of the BS group lost a significant amount of weight (>7%). The main causes for the dropouts were the lack of interest to continue (71%) in the LS group, and living outside Riyadh (61%) in the BS group. 

### 3.3. Changes in the Body Composition

A comparison of the baseline and final assessments regarding body composition parameters is shown in Table 2. In the BS group, there were significant reductions in PBF, FM, FMI, and visceral fat rating compared with other groups. The fat-free mass and TBW were insignificantly different among groups, indicating that the BS mainly induces loss of body adiposity. The before-after comparison showed significant changes in both BS and WL. However, the WR showed insignificant body compositional changes. The weight and fat mass changes are presented in Figure 2. The BS group showed superior results versus both the WL and WR groups.

### 3.4. Changes in the Inflammatory Cytokines

A comparison of the baseline and final assessments regarding the studied inflammatory cytokines showed significant differences in the BS and WL groups (Table 3). In the WR group, significant differences were found in levels of CRP and MCP-1. A comparison of the three groups at the final assessment showed insignificant differences.

### 3.5. Changes in the Antioxidants Activity and Oxidative Stress Markers

As shown in Table 4, antioxidant enzyme activities, glutathiones, and the oxidative stress parameters were significantly improved in the BS group 6 months after surgery. In the WL group, most of the measured parameters were improved. Interestingly, despite insignificant weight loss in the WR group, some antioxidant enzymes (GPx, GR, GST, GS, and SOD) were significantly increased and some reactive oxygen species (superoxide anions, lipid peroxides, and peroxynitrite radicals) were significantly reduced. Apart from superoxide anions, all the parameters were significantly improved in the BS when comparing the final assessment values among the three groups.

### 3.6. Changes in the VCD Risk

In the BS group, LDL cholesterol, FRS-BMI-Hard CVD, and FRS-BMI-Full CVD were significantly reduced after 6 months. In the WL group, only the FRS-BMI-Hard CVD, and FRS-BMI-Full CVD showed significant reductions. In the WR group, no significant changes were reported (Table 5).

### 3.7. Correlation of Weight Loss with Study Parameters 

The correlations of weight loss amount and fat mass change with the concentrations of inflammatory markers and levels of the CVD risk by the FRS and ASCVD at the final assessment are shown in Table 6. The amount of weight loss failed to show any significant correlation with the inflammatory cytokines or CVD risk in the BS while the fat mass loss showed significant inverse correlations with the CVD risk by FRS and ASCVD. In the WL group, both weight loss and fat mass loss were inversely associated with the levels of CRP and MCP1. In addition, fat mass loss showed an inverse correlation with the ASCVD. In the WR group, the changes in weight and fat mass were mainly negative (i.e., increased at the time of final assessment) so some positive correlations were shown with the CVD risk scores. 

## 4. Discussion

This study compared the body composition changes, improvements in the inflammatory status, changes in the antioxidant enzymes activity, and oxidative stress parameters before and after 6 months of intervention for obesity management by SG or LS with or without successful weight loss. The first main finding is the reduction in the serum levels of inflammatory cytokines in the BS group. The remission of CVD risk and other comorbidities of obesity is linked to the resolution of obesity-induced inflammation [10]. Levels of MCP-1 were reduced significantly after procedures at different postoperative periods [23,24] and showed insignificant changes in another study [25]. This ameliorative effect was also reported in levels of IL-6, TNF-α, and CRP [26]. TNF-α was the most inconsistent cytokine after BS in the literature. Some studies reported insignificant changes after RYGB [27] and after gastric banding [28], whereas other studies reported an increased level of TNF-α three months after RYGB [29]. The discrepancy may be caused by differences in the baseline comorbidities, different surgical procedures, timing of assessment during the postoperative period, and variability in the measuring techniques for inflammatory mediators. The inflammatory amelioration comes after and may be caused by an improvement in the metabolic status after the BS [30]. It was reported that the cell composition of the immune system can recover from the obesity-associated detrimental effects within three months after the BS; however, the recovery of normal cytokine release may take a longer time [31].

Lifestyle intervention with multiple techniques including diet, physical activity, and behavioral modification produced reductions in the measured cytokines. This was consistent with a longer (1 year) trial of a multidisciplinary weight reduction program in women with obesity which reported a modest weight loss and significant reductions in TNF-α and IL-6 as well as significant correlations between sustained weight loss and reductions in cytokines levels, adhesin concentrations and the improvement of vascular responses to l-arginine [32]. Patients with obesity and type 2 diabetes showed significant reductions in TNF-α, IL-6, CRP, conjugated dienes, and MDA in addition to significant rises in GPx, SOD, and GSH after three months of a lifestyle intervention (diet and physical activity) [33]. Similar results were reported in patients with metabolic syndrome [13,14]. Notably, our participants were experiencing a more healthy obesity, i.e., free of diabetes or metabolic syndrome. Interestingly, the participants of the WR group showed significant reductions in levels of CRP and MCP1 by lifestyle changes without significant weight loss. This was consistent with a German study where healthy lifestyle changes can lower levels of CRP and adiponectin in an overweight general population [34].

The second main finding in this study was the amelioration of the oxidative stress markers and the increase in antioxidant enzyme activities after weight loss by SG and LS. Additionally, despite insignificant weight loss in the WR group, some improvement was detected. It was reported that weight loss, either through LS or BS, ameliorates OS and increased antioxidant enzymes in patients with obesity [35]. Metere et al. [36] found that SG could normalize the reactive oxygen species, as measured by electron paramagnetic resonance spectroscopy, 6 to 12 months after surgery. Furthermore, Carmona-Maurici et al. [37] reported that BS reduced OS and improved the antioxidant profile (measured by ELISA) in patients with and without atheromatous plaques 6 months after surgery. In another study, levels of total antioxidant capacity, reduced glutathione, MDA, and CRP were reduced 4 months after RYGB surgery [12]. Regarding LS trials, Monserrat-Mesquida [16] reported that a two-year nutritional and lifestyle intervention could reduce oxidative and inflammatory status, resulting in a reduction in CVD risk in individuals with high CVD risk. On the other hand, LS interventions did not produce any amelioration in the OS in women with obesity and infertility with modest weight loss [15].

The current study showed a significant reduction in CVD risk identified by FRS-BMI for both full and hard cardiovascular events in both BS and LS, whereas the ASCVD and FRS-Lipid-Hard and full CVD showed insignificant reductions. Moreover, fat mass loss correlated significantly with FRS-BMI and ASCVD in the BS group, while in the WL group fat mass loss correlated only with ASCVD. A study with a large cohort of 1048 patients undergoing RYGB showed a significantly lower CVD risk from the first postoperative year and was sustained for five years after surgery [38]. Blanco et al. [39] performed the ASCVD risk score and the FRS retrospectively for patients who underwent SG and RYGB and found that both procedures were equally effective in significantly reducing CVD risk. In another study, SG was prospectively compared with RYGB for 12 months. The ASCVD scores showed more risk reduction in the SG patients than in the RYGB patients [40]. For non-surgical weight loss methods, the effect of a two-year weight reduction trial on the seven-year risk of CVD showed that the weight loss produced a lower risk of coronary heart disease but not for stroke events [41].

The link between inflammation, OS, and CVD risk during weight loss is clear in the current study. The improvement in the CVD risk might be a result of the reduction in inflammatory cytokines, relief of oxidative stress, and enhancement of antioxidant enzyme activities. In another study, twenty men with obesity after 12 weeks of lifestyle modifications showed reductions in BMI, cytokines levels, the brachial-ankle pulse wave velocity (as an index of arterial stiffness), and a positive correlation between changes in the brachial-ankle pulse wave velocity and circulating IL-6 levels. They suggested that LS intervention reduced arterial stiffness by reducing IL-6 levels [42]. The current study showed that fat mass reduction produced more significant correlations with CVD risk scores than body weight loss. This was consistent with the finding of Serin et al. [43] who found that sustained fat mass loss rather than body weight loss can reduce the state of low-grade inflammation and induce promotion of the cardiometabolic health in the normal-weight population. A healthy lifestyle could lower systemic inflammation and potentially lower the CVD risk, even in patients with established cardiovascular disease [44].

Taken together, our findings suggest that successful lifestyle modification could produce a promising amelioration of inflammatory status, oxidative stress, and CVD risk with a degree similar to BS but without surgical stress and possible perioperative complications. This benefit is associated with the reduction in fat mass rather than weight loss. Cases with morbid obesity who are considered candidates for the BS should start a period of multimodal LS intervention before surgery for maximum cardiometabolic benefits and better inflammatory outcomes.

Despite this article having some strengths, such as comparisons of BS and LS with and without significant weight loss in a prospective manner, this study encounters some limitations. The main limitation was the relatively high dropout rate. The high dropout rate is frequent in our locality, especially with prospective studies for obesity management over a long period [45]. Another limitation was using the CVD risk algorithms rather than objective methods of CVD assessment. The third limitation was the simple measurements at baseline and at 6 months which might mask some intermediate benefits. 

## 5. Conclusions

In conclusion, BS produced superior weight and fat mass loss. However, both BS and LS produced a relatively similar reduction in the inflammatory cytokines, relief of oxidative stress indicators, and enhancement of antioxidant capacity, and consequently may reduce the CVD risk algorithm. These beneficial effects are related to the loss of fat mass rather than body weight loss.

## Figures and Tables

**Figure 1 medicina-59-00751-f001:**
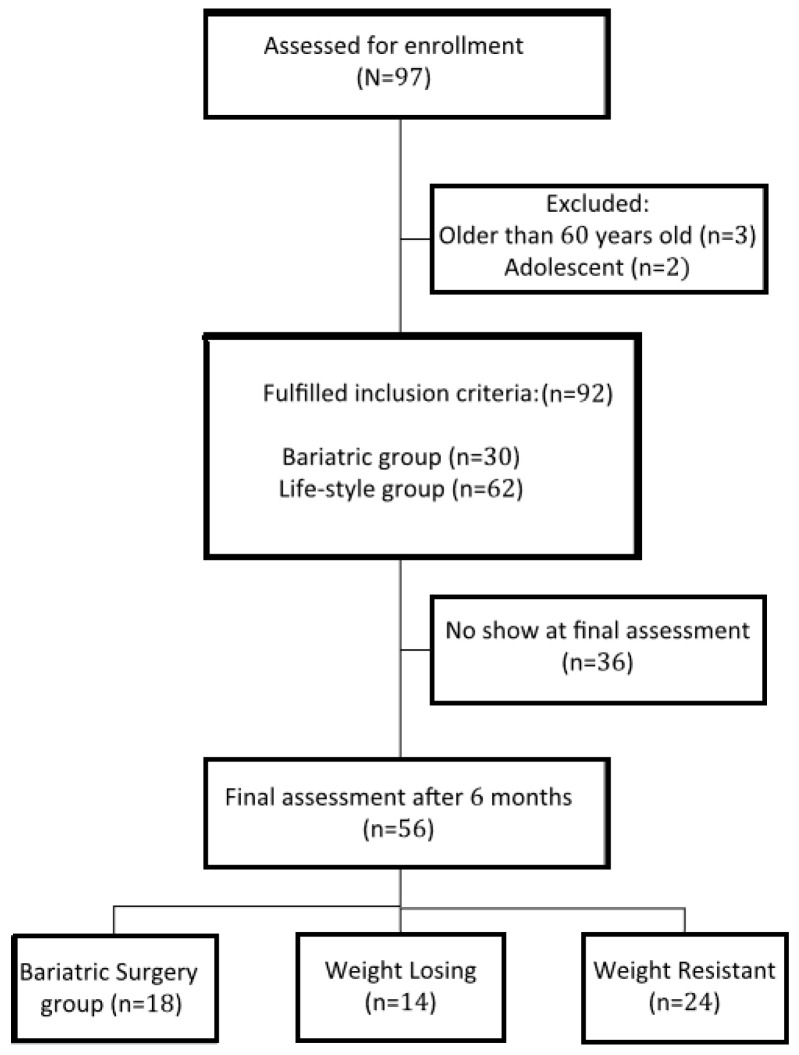
Flow diagram of the progress through the phases of the study.

**Figure 2 medicina-59-00751-f002:**
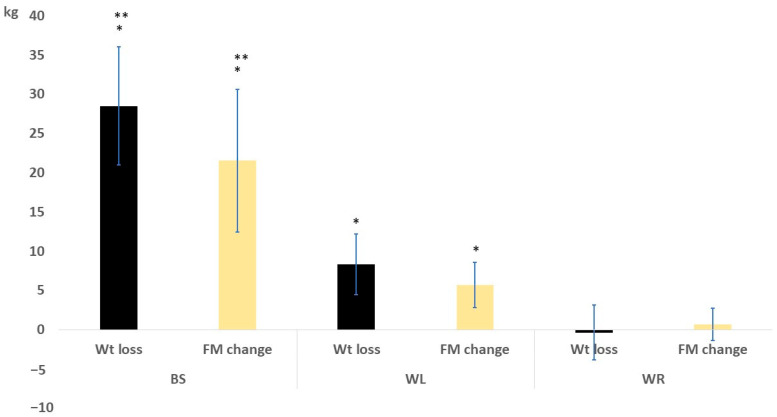
Weight and fat mass change after 6 months in the study groups. * means significant vs the WR group; ** means significant vs the WL group.

**Table 1 medicina-59-00751-t001:** Baseline data of all study groups.

Variables	BS Group Mean ± SD (n = 30)	LS Group Mean ± SD (n = 62)	*p*-Value
Age (years)	31.67 ± 12.93	37.10 ± 12.84	0.306
Weight (kg)	115.61 ± 18.49	113.66 ± 17.42	0.788
Height (cm)	165.22 ± 8.45	167.47 ± 8.84	0.529
BMI (kg/m^2^)	42.18 ± 4.62	40.29 ± 3.31	0.227
WC (cm)	120.71 ± 13.87	107.71 ± 15.38	0.123
HC (cm)	133.14 ± 10.81	130.36 ± 9.84	0.623

**Table 2 medicina-59-00751-t002:** Changes in body composition among study groups.

Variables	BS (n = 18)			WL (n = 14)			WR (n = 24)			*p*-Value *
	Baseline	Final	*p*-Value	Baseline	Final	*p*-Value	Baseline	Final	*p*-Value	
PBF	44.14 ± 6.80	33.86 ± 7.91 ^b^	0.001	40.26 ± 7.49	37.86 ± 8.66 ^a^	0.013	43.58 ± 6.60	43.41 ± 6.15 ^a^	0.623	0.028
Fat mass (kg)	51.11 ± 11.94	29.59 ± 8.59 ^c^	<0.001	45.17 ± 10.45	39.47 ± 10.92 ^b^	0.002	48.11 ± 6.51	47.46 ± 6.70 ^a^	0.322	<0.001
Fat mass index (kg/m^2^)	18.77 ± 4.33	10.92 ± 3.42 ^b^	<0.001	16.09 ± 3.66	14.10 ± 4.02^a^	0.001	17.47 ± 2.76	17.02 ± 2.62 ^a^	0.187	0.001
Fat-free mass (kg)	64.50 ± 12.48	57.54 ± 11.02 ^a^	0.003	67.37 ± 13.94	64.76 ± 12.62 ^a^	0.008	63.61 ± 15.03	64.09 ± 16.03 ^a^	0.507	0.485
Fat-free mass index (kg/m^2^)	23.41 ± 2.63	20.86 ± 1.93 ^a^	0.003	23.69 ± 3.01	22.80 ± 2.66 ^a^	0.005	22.67 ± 3.42	22.53 ± 3.39 ^a^	0.718	0.284
Muscle mass (kg)	61.37 ± 11.94	54.80 ± 10.66 ^a^	0.004	64.16 ± 13.39	61.70 ± 12.15 ^a^	0.008	60.54 ± 14.42	60.97 ± 15.37 ^a^	0.521	0.493
Visceral fat rate	12.78 ± 8.24	9.44 ± 4.39 ^b^	0.203	17.43 ± 4.93	15.00 ± 5.03 ^a^	0.002	15.55 ± 4.97	15.82 ± 5.02 ^a^	0.341	0.018
Total body water (kg)	47.22 ± 9.13	42.12 ± 8.08 ^a^	0.003	49.31 ± 10.20	47.40 ± 9.24 ^a^	0.007	46.57 ± 10.99	46.91 ± 11.73 ^a^	0.532	0.485

* *p*-value among values at the final assessment. PBF means percent body fat. Superscripts (a, b, and c) with different values indicate significantly different values at the final assessment for the intergroup comparisons (the reference control group was the WR group).

**Table 3 medicina-59-00751-t003:** Changes in inflammatory cytokines among study groups.

Variables	BS (n = 18)			WL (n = 14)			WR (n = 24)			*p*-Value *
	Baseline	Final	*p*-Value	Baseline	Final	*p*-Value	Baseline	Final	*p*-Value	
C reactive protein (mg/dL)	84.98 ± 2.94	62.12 ± 5.86 ^a^	<0.001	84.56 ± 3.58	62.91 ± 3.09 ^a^	<0.001	79.07 ± 6.93	58.57 ± 5.00 ^a^	<0.001	0.279
IL-6 (pg/mL)	20.64 ± 5.00	11.00 ± 1.22 ^a^	<0.001	18.42 ± 4.89	10.58 ± 2.50 ^a^	0.001	14.00 ± 4.32	10.29 ± 3.15 ^a^	0.129	0.832
TNF-α (pg/mL)	46.56 ± 3.64	16.22 ± 2.54 ^a^	<0.001	34.42 ± 14.18	18.42 ± 5.16 ^a^	0.008	26.57 ± 13.50	16.71 ± 5.19 ^a^	0.135	0.512
MCP-1 (pg/mL)	646.11 ± 55.59	278.67 ± 50.98 ^a^	<0.001	473.92 ± 147.56	270.08 ± 62.50 ^a^	0.001	472.86 ± 180.31	240.57 ± 46.34 ^a^	0.009	0.380

* *p*-value among values at the final assessment. Superscripts (such as a) with different values indicate significantly different values at the final assessment for the intergroup comparisons (the reference control group was the WR group).

**Table 4 medicina-59-00751-t004:** Changes in antioxidant enzyme activities and oxidative stress parameters among study groups.

Variables	BS (n = 18)			WL (n = 14)			WR (n = 24)			*p*-Value *
	Baseline	Final	*p*-Value	Baseline	Final	*p*-Value	Baseline	Final	*p*-Value	
Antioxidant Enzyme Activity										
GPx (nmole/min/mL)	1.89 ± 0.09	2.36 ± 0.12 ^b^	<0.001	2.14 ± 0.29	2.40 ± 0.44 ^b^	0.058	1.86 ± 0.15	2.03 ± 0.16 ^a^	0.038	0.028
GR (nmole/min/mL)	1.30 ± 0.09	1.59 ± 0.08 ^b^	0.001	1.11 ± 0.17	1.35 ± 0.21 ^b^	0.010	1.04 ± 0.10	1.19 ± 0.09 ^a^	0.008	<0.001
GST (pmole/min/mL)	77.19 ± 5.73	88.08 ± 4.28 ^b^	0.008	58.42 ± 7.31	65.88 ± 10.25 ^a^	0.033	52.30 ± 5.93	60.54 ± 6.95 ^a^	0.028	<0.001
GCL (pmol/min/mL)	528.20 ± 24.46	598.23 ± 15.25 ^c^	<0.001	494.14 ± 49.51	541.85 ± 63.99 ^b^	0.010	479.80 ± 29.55	500.16 ± 32.58 ^a^	0.345	0.001
GS (nmole/min/mL)	2.75 ± 0.21	3.34 ± 0.30 ^a^	<0.001	3.15 ± 0.35	3.57 ± 0.47 ^b^	0.076	2.69 ± 0.19	3.16 ± 0.35 ^a^	0.012	0.111
SOD (nmole/min/mL)	4.09 ± 0.24	4.96 ± 0.16 ^c^	<0.001	3.62 ± 0.32	4.43 ± 0.52 ^b^	0.001	3.17 ± 0.31	3.55 ± 0.43 ^a^	0.043	<0.001
CAT (nmole/min/mL)	3.65 ± 0.41	5.46 ± 0.70 ^c^	0.002	3.71 ± 0.28	4.36 ± 0.38 ^b^	0.006	3.52 ± 0.38	3.82 ± 0.44 ^a^	0.293	<0.001
Reduced/oxidized glutathione										
GSH (umole/L)	2.13 ± 0.24	3.01 ± 0.27 ^b^	<0.001	2.66 ± 0.27	3.10 ± 0.32 ^b^	0.005	2.59 ± 0.11	2.58 ± 0.31 ^a^	0.983	0.005
GSSG (umole/L)	0.119 ± 0.014	0.093 ± 0.008 ^b^	<0.001	0.119 ± 0.014	0.097 ± 0.010 ^b^	0.006	0.131 ± 0.008	0.121 ± 0.005 ^a^	0.001	<0.001
GSH/GSSG Ratio	18.15 ± 3.11	32.34 ± 1.06 ^a^	<0.001	22.60 ± 3.55	32.52 ± 5.26 ^a^	<0.001	19.72 ± 1.19	21.30 ± 2.40 ^a^	0.149	<0.001
Reactive oxygen species										
Hydrogen Peroxide (nmole/mL)	12.42 ± 0.64	10.90 ± 0.52 ^b^	<0.001	12.29 ± 0.91	11.57 ± 0.59 ^a^	0.137	12.47 ± 0.68	12.03 ± 0.98 ^a^	0.292	0.017
Superoxide Anion (nmol/mL)	55.66 ± 4.26	43.87 ± 3.46 ^a^	0.001	51.26 ± 5.60	43.70 ± 3.69 ^a^	0.010	59.58 ± 5.18	48.61 ± 8.71 ^a^	0.002	0.184
Lipid Peroxides (nmole/mL)	6.98 ± 0.24	6.04 ± 0.28 ^c^	<0.001	7.96 ± 0.67	7.04 ± 0.55 ^b^	0.009	8.38 ± 0.35	7.85 ± 0.24 ^a^	0.012	<0.001
t-protein carbonyl content (pmole/mL)	334.18 ± 25.58	262.53 ± 19.59 ^c^	<0.001	348.74 ± 11.68	306.08 ± 29.20 ^b^	0.010	348.95 ± 14.26	331.21 ± 22.61 ^a^	0.185	<0.001
Peroxynitrite radical (µmole/mL)	5.40 ± 0.73	4.29 ± 0.37 ^c^	0.015	6.02 ± 0.29	5.48 ± 0.43 ^b^	0.012	6.72 ± 0.29	6.28 ± 0.25 ^a^	0.004	<0.001
Nitic oxide (µmole/mL)	64.73 ± 6.29	52.10 ± 3.85 ^b^	<0.001	67.10 ± 7.39	55.44 ± 3.44 ^b^	0.004	71.71 ± 9.06	71.16 ± 5.70 ^a^	0.885	<0.001

GPx = Glutathione peroxidase; GR = Glutathione reductase; GST = Glutathione S-transferase; GCL = Glutamate cysteine ligase; GS = Glutathione synthetase; SOD = Superoxide dismutase; CAT = Catalase; GSH = reduced glutathione; GSSG = oxidized glutathione. * *p*-value among values at the final assessment. PBF means percent body fat. Superscripts (a, b, and c) with different values indicate significantly different values at the final assessment for the intergroup comparisons (the reference control group was the WR group).

**Table 5 medicina-59-00751-t005:** Changes in CVD risk among study groups.

Variables	BS (n = 18)			WL (n = 14)			WR (n = 24)			*p*-Value *
	Baseline	Final	*p*-Value	Baseline	Final	*p*-Value	Baseline	Final	*p*-Value	
TCholesterol (mg/dL)	186.57 ± 38.89	194.82 ± 50.87 ^a^	0.466	156.62 ± 38.73	164.83 ± 36.73 ^a^	0.847	169.86 ± 55.86	187.46 ± 52.50 ^a^	0.397	0.827
HDL(mg/dL)	43.41 ± 7.61	57.24 ± 43.34 ^a^	0.370	36.65 ± 8.18	33.20 ± 6.04 ^a^	0.676	38.06 ± 17.91	36.74 ± 9.37 ^a^	0.193	0.494
LDL(mg/dL)	124.97 ± 32.68	105.73 ± 59.88 ^b^	0.006	99.40 ± 28.54	115.94 ± 32.66 ^a^	0.533	112.38 ± 52.52	132.81 ± 45.62 ^a^	0.927	0.164
TG(mg/dL)	92.25 ± 37.84	68.89 ± 21.71 ^b^	0.593	102.14 ± 83.45	84.01 ± 44.65 ^b^	0.088	97.09 ± 59.15	108.13 ± 35.09 ^a^	0.193	0.521
Fasting glucose (mg/dL)	95.82 ± 40.43	88.11 ± 30.90 ^a^	0.421	73.85 ± 14.32	79.45 ± 22.82 ^a^	0.930	91.20 ± 15.98	84.43 ± 18.96 ^a^	0.365	0.084
FRS-BMI-Hard CVD	16.89 ± 24.38	13.78 ± 23.60 ^a^	0.003	23.43 ± 16.79	20.00 ± 15.09 ^a^	0.007	22.00 ± 22.32	21.33 ± 21.36 ^a^	0.296	0.701
FRS-BMI-Full CVD	24.22 ± 26.78	19.89 ± 26.26 ^a^	0.004	35.43 ± 21.88	31.00 ± 19.85 ^a^	0.009	31.75 ± 25.58	31.08 ± 24.88 ^a^	0.266	0.532
FRS-Lipid-Hard CVD	10.00 ± 14.32	11.44 ± 18.19 ^a^	0.356	13.57 ± 8.62	13.43 ± 10.97 ^a^	0.929	10.00 ± 14.32	14.42 ± 11.52 ^a^	0.348	0.889
FRS-Lipid-Full CVD	16.67 ± 18.71	18.00 ± 22.92 ^a^	0.068	24.71 ± 14.66	24.57 ± 16.61 ^a^	0.939	22.75 ± 18.00	24.92 ± 18.09 ^a^	0.312	0.693
ASCVD	43.89 ± 11.88	38.56 ± 18.21 ^a^	0.179	40.00 ± 7.90	36.29 ± 7.76 ^a^	0.254	39.83 ± 8.03	38.33 ± 6.93 ^a^	0.231	0.918

* *p*-value among values at the final assessment. Superscripts (a, b) with different values indicate significantly different values at the final assessment for the intergroup comparisons (the reference control group was the WR group).

**Table 6 medicina-59-00751-t006:** The correlation of weight loss with other study variables in study groups.

Variables at the Final Assessment	Weight Loss			Fat Mass Loss		
	BS Group (n = 18)	WL Group (n = 14)	WR Group (n = 24)	BS Group (n = 18)	WL Group (n = 14)	WR Group (n = 24)
CRP	0.357	−0.571 *	0.600 *	0.143	−0.750 **	0.400
IL6	0.298	−0.148	−0.033	0.298	−0.259	0.047
TNFa	−0.188	−0.107	−0.089	−0.214	−0.071	0.164
MCP1	0.218	−0.786 **	−0.077	0.460	−0.750 **	0.369
FRS-BMI-Hard CVD	−0.310	−0.107	0.427	−0.619 *	−0.071	0.745 **
FRS-BMI-Full CVD	−0.310	−0.107	0.441	−0.619 *	−0.071	0.727 **
FRS-Lipid-Hard CVD	0.390	0.429	0.291	−0.661 *	−0.321	0.601 *
FRS-Lipid-Full CVD	0.400	0.429	0.414	−0.583 *	−0.321	0.579 *
ASCVD	0.380	−0.472	0.130	−0.759 *	−0.567 *	0.651 *

* Correlation is significant at the 0.05 level (2-tailed); ** Correlation is significant at the 0.01 level (2-tailed).

## Data Availability

The datasets for this study are part of an ongoing project. However, they can be requested from the PI without undue reservation to any qualified researcher.

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
