# Peer review of "Inflammatory Cytokines, Redox Status, and Cardiovascular Diseases Risk after Weight Loss via Bariatric Surgery and Lifestyle Intervention"

_medicina, 2023, doi:10.3390/medicina59040751_

Round 1

Reviewer 1 Report

This is a nice study in which the authors find that despite the superior weight loss achieved with BS, the impact of effective adoption of diet and exercise and behavioral modification (LS-WL) induces similar reductions in the inflammatory cytokines, oxidative stress indicators, and enhancement of antioxidant capacity, and CVD risk algorithm reduction.

Please find below my questions, comments and suggestions.

The main indication for BS is  on patients with a BMI of 40 kg/m2 or greater without coexisting medical problems. In addition, patients with a BMI ≥ 35 kg/m2 and 1 or more severe obesity-related comorbidities, including type 2 diabetes, hypertension, hyperlipidemia, obstructive sleep apnea (OSA), non-alcoholic fatty liver disease etc, would qualify as surgical candidates.

1.- On what basis was the inclusion criteria of BMI ≥35 kg/m2 selected?

2.-  24-25 Measurements were taken before and after six months of either SG or LS (500-kcal-deficit balanced diet.

112-114  For participants in the lifestyle group, a baseline assessment was done at the first 112 visit, then every patient was given a balanced 500 to 1000-calorie-deficit diet, personalized 113 physical activity plan, and customized behavioral modification according to our previous 114 publication [17].

It is not clear if both groups followed or not 500 or 500 to 1000-calorie-deficit diet

3.- Table 1. Baseline data of all study groups.

BMI (kg/m²)

42.18±4.62a

40.29±3.31

0.227

The meaning of the superscript "a" in the table is not clear. It is suggested to describe it at the bottom of the table or remove it if redundant.

4.- Table 2: Changes in body composition among study groups.

-191-192 * p-value among values at the final assessment. PBF means percent body fat. Different superscripts (a, b, and c) indicate significantly different.

I assume that the superscripts a, b and c correspond to the paired comparisons of the BS, WL and WR groups respectively. However, it is not clear enough, especially when a appears, for example, in the BS group, in addition to the corresponding value of P, before and after, in each group. It is suggested to be more explicit at the bottom of the table.

The same for the rest of the tables

5.- Given the risk involved in a BS compared to adopting LS and taking into account the similar benefits of both interventions, I think it would be reasonable to briefly include a comment on that in the discussion section.

6.- Finally, a comment and a question

It is surprising the benefits obtained in the three groups, including those in the WR group. In my opinion, an additional limitation of the study is  that simple measurements of all variables at baseline and at 6 months could mask some of the benefits that could be achieved in the interim. Thus, the changes in body weight observed in an intervention are highly variable in a group and even in the same individual. Wouldn't it have been better to establish the body change of the study subjects as the area under the curve from 0 to 6 months instead of two simple measurements 0 and 6 months?

Author Response

Response to Reviewer 1

Manuscript ID:  medicina-2304123

Manuscript Title: “Inflammatory Cytokines, redox status, and cardiovascular diseases risk after weight loss via bariatric surgery and lifestyle intervention”

We thank the reviewers for their careful examination of the manuscript and appreciate the useful suggestions to improve the quality of our paper. Our point-by-point response to the reviewers' comments is given below. Changes in the manuscript are indicated in red font. Please note that the pages and line numbers mentioned in the reviewers’ comments refer to the original manuscript, whereas those in the authors’ reply refer to the revised manuscript.

Comments from the Editors and Reviewers:

  1. This is a nice study in which the authors find that despite the superior weight loss achieved with BS, the impact of effective adoption of diet and exercise and behavioral modification (LS-WL) induces similar reductions in the inflammatory cytokines, oxidative stress indicators, and enhancement of antioxidant capacity, and CVD risk algorithm reduction.

Answer: Thank you for pointing this out.

  1. The main indication for BS is in patients with a BMI of 40 kg/m2 or greater without coexisting medical problems. In addition, patients with a BMI ≥ 35 kg/m2 and 1 or more severe obesity-related comorbidities, including type 2 diabetes, hypertension, hyperlipidemia, obstructive sleep apnea (OSA), non-alcoholic fatty liver disease etc, would qualify as surgical candidates..

1.- On what basis were the inclusion criteria of BMI ≥35 kg/m2 selected?

Answer: according to the 2022 American Society for Metabolic and Bariatric Surgery (ASMBS) and International Federation for the Surgery of Obesity and Metabolic Disorders (IFSO) guidelines, a body mass index (BMI) ≥40 kg/m2, or BMI ≥35 kg/m2 with co-morbidities, is a threshold for surgery that is applied universally (DOI: https://doi.org/10.1016/j.soard.2022.08.013). The Mean±SD of our sample BMI was 42.18±4.62 kg/m2. The participants below 40 were only two in number with BMI 35.5 and 36.2 respectively and both of them were suffering from hypertension and diabetes.

  1. - Measurements were taken before and after six months of either SG or LS (500-kcal-deficit balanced diet.

12-114  For participants in the lifestyle group, a baseline assessment was done at the first visit, then every patient was given a balanced 500 to 1000-calorie-deficit diet, personalized physical activity plan, and customized behavioral modification according to our previous publication [17].

It is not clear if both groups followed or not 500 or 500 to 1000-calorie-deficit diet

Answer: The 500 to 1000-calorie-deficit diet was only for the LS group. While the BS received the traditional postoperative dietary management (which is not a 500 to 1000-calorie-deficit diet). This is clear now in the 2.2 section.

  1. - Table 1. Baseline data of all study groups.

            BMI (kg/m²)

42.18±4.62a

40.29±3.31

0.227

The meaning of the superscript "a" in the table is not clear. It is suggested to describe it at the bottom of the table or remove it if redundant.

Answer: Sorry for this oversight. it was a typo mistake and it is now deleted.

  1. - Table 2: Changes in body composition among study groups.

-191-192 * p-value among values at the final assessment. PBF means percent body fat. Different superscripts (a, b, and c) indicate significantly different.

I assume that the superscripts a, b, and c correspond to the paired comparisons of the BS, WL, and WR groups respectively. However, it is not clear enough, especially when a appears, for example, in the BS group, in addition to the corresponding value of P, before and after, in each group. It is suggested to be more explicit at the bottom of the table.

The same for the rest of the tables.

Answer: sorry for this oversight. It is now clarified in the footnote of the tables.

  1. - Given the risk involved in a BS compared to adopting LS and taking into account the similar benefits of both interventions, I think it would be reasonable to briefly include a comment on that in the discussion section.

Answer: Thanks for this suggestion. We added a paragraph in the discussion at lines 338-344.

  1. - Finally, a comment and a question

It is surprising the benefits obtained in the three groups, including those in the WR group. In my opinion, an additional limitation of the study is that simple measurements of all variables at baseline and at 6 months could mask some of the benefits that could be achieved in the interim. Thus, the changes in body weight observed in an intervention are highly variable in a group and even in the same individual. Wouldn't it have been better to establish the body change of the study subjects as the area under the curve from 0 to 6 months instead of two simple measurements 0 and 6 months?

Answer: Thanks for this suggestion. This limitation was added. The use of multiple points of time for assessment was costly and some patients especially BS participants cannot tolerate frequent appointments, especially those coming from outside Riyadh city.

Reviewer 2 Report

Review on manuscript titled “Inflammatory cytokines, redox status, and cardiovascular dis-2 eases risk after weight loss via bariatric surgery and lifestyle 3 intervention” submitted to Medicina for publication

General remark

The paper presents an effort to analyse the weight loss after bariatric surgery and lifestyle changes in obese patients. The topic is of high importance since obesity, its possible treatment and comorbidities are in the focus of medical, epidemiological, health economic studies. The methods of the analysis are appropriate.

I list my specific/minor comments and suggestions to the manuscript in the order of the chapters of the manuscript:

Abstract

A1: “The LS group was 20 subdivided into the weight loss (WL) and weight resistance (WR) groups based on the achievement 21 of 7% weight loss.” – why only LS group was divided into subgroups, why not the BS group either? This should be mentioned in the Abstract, too.

Introduction

I1: “Lifestyle intervention procedures are less invasive …” – please correct this sentence, lifestyle intervention is not invasive.

Material and methods

M1: What kind of behavioral modifications was suggested / used in the LS group beside the multiple techniques including diet, physical activity changes? Please give more details on diet, physical activity and behavioral modifications in LS group during the intervention period.

Results

R1: “At the final assessment, committed participants in the LS group were divided into 175 WL (n=14, 42.8% women) and WR (n=24, 41.7% women). The remaining were dropped 176 out (n=24). The BS group also had a no-show for 12 participants.” - does this mean that no one in the BS group lost less than 7% of baseline weight? That is why there are not WL and WR subgroups in BS group?

R2: “However, the WR showed significant body compositional changes.” – what kind of changes were found in this group? Please, complete the Results section with these changes.

R3: What are the supposed causes of fat free mass/index decreases in the BS, WL, WR groups during the studied period?

R3: “A comparison of the three groups at the final assessment showed insignificant differences.” – p values in Table 3 do not represent significant changes (p: 0.279, 0.832, 0.512, 0.380), please revise this sentence.

Figures

Figure 2: The dimension (kg) of weight and fat mass is missing from the figure/legend, please compete the figure with the dimension.

Author Response

Response to Reviewer 2

Manuscript ID:  medicina-2304123

Manuscript Title: “Inflammatory Cytokines, redox status, and cardiovascular diseases risk after weight loss via bariatric surgery and lifestyle intervention”

We thank the reviewers for their careful examination of the manuscript and appreciate the useful suggestions to improve the quality of our paper. Our point-by-point response to the reviewers' comments is given below. Changes in the manuscript are indicated in red font. Please note that the pages and line numbers mentioned in the reviewers’ comments refer to the original manuscript, whereas those in the authors’ reply refer to the revised manuscript.

Comments from the Editors and Reviewers:

  1. General remark

The paper presents an effort to analyze the weight loss after bariatric surgery and lifestyle changes in obese patients. The topic is of high importance since obesity, its possible treatment, and comorbidities are in the focus of medical, epidemiological, health economic studies. The methods of the analysis are appropriate.

I list my specific/minor comments and suggestions to the manuscript in the order of the chapters of the manuscript:

Answer: Thank you for the perfect summarization

  1. Abstract

A1: “The LS group was 20 subdivided into the weight loss (WL) and weight resistance (WR) groups based on the achievement 21 of 7% weight loss.” – why only LS group was divided into subgroups, why not the BS group either? This should be mentioned in the Abstract, too.

Answer: Thanks for this comment. This was mentioned in lines 20-22. All participants in the BS group had significant weight reduction > 7%. So the division of the BS will be not logical. The idea here is to have a control group with insignificant weight loss which was the WR group as they failed to achieve a significant weight loss or even gain weight.

  1. Introduction

I1: “Lifestyle intervention procedures are less invasive …” – please correct this sentence, lifestyle intervention is not invasive.

Answer: thanks for this comment. It is corrected.

  1. Material and methods

M1: What kind of behavioral modifications was suggested / used in the LS group beside the multiple techniques including diet, physical activity changes? Please give more details on diet, physical activity and behavioral modifications in LS group during the intervention period.

Answer: Thanks for this advice we added more details in lines 117-130.

  1. Results

R1: “At the final assessment, committed participants in the LS group were divided into 175 WL (n=14, 42.8% women) and WR (n=24, 41.7% women). The remaining were dropped 176 out (n=24). The BS group also had a no-show for 12 participants.” - does this mean that no one in the BS group lost less than 7% of baseline weight? That is why there are not WL and WR subgroups in BS group?

Answer: All attendant participants of the BS group lost a significant amount of weight (>7%). As reported In figure 2 mean±SD of weight loss in the BS group was 28.48±7.54 kg.

  1. R2: “However, the WR showed significant body compositional changes.” – what kind of changes were found in this group? Please, complete the Results section with these changes.

Answer: Sorry for this typo mistake. It is “insignificant”.

  1. R3: What are the supposed causes of fat-free mass/index decreases in the BS, WL, and WR groups during the studied period?

Answer: It is usually due to loss of body water and muscle mass, especially in those who didn’t practice a considerable periods of physical activity.

  1. R3: “A comparison of the three groups at the final assessment showed insignificant” – p values in Table 3 do not represent significant changes (p: 0.279, 0.832, 0.512, 0.380), please revise this sentence.

Answer: Yes all are insignificant at the final assessment.

  1. Figures

Figure 2: The dimension (kg) of weight and fat mass is missing from the figure/legend, please complete the figure with the dimension.

Answer: Added, thanks
